# Fast and Economic Microarray-Based Detection of Species-, Resistance-, and Virulence-Associated Genes in Clinical Strains of Vancomycin-Resistant Enterococci (VRE)

**DOI:** 10.3390/s24196476

**Published:** 2024-10-08

**Authors:** Ibukun Elizabeth Osadare, Stefan Monecke, Abdinasir Abdilahi, Elke Müller, Maximilian Collatz, Sascha Braun, Annett Reissig, Wulf Schneider-Brachert, Bärbel Kieninger, Anja Eichner, Anca Rath, Jürgen Fritsch, Dominik Gary, Katrin Frankenfeld, Thomas Wellhöfer, Ralf Ehricht

**Affiliations:** 1Leibniz Institute of Photonic Technology (IPHT), Research Alliance Leibniz Center for Photonics in Infection Research (LPI), 07745 Jena, Germany; ibukun-elizabeth.osadare@leibniz-ipht.de (I.E.O.); stefan.monecke@leibniz-ipht.de (S.M.); abdinasir.abdilahi@uni-jena.de (A.A.); elke.mueller@leibniz-ipht.de (E.M.); maximilian.collatz@leibniz-ipht.de (M.C.); sascha.braun@leibniz-ipht.de (S.B.); annett.reissig@leibniz-ipht.de (A.R.); 2InfectoGnostics Research Campus, 07743 Jena, Germany; 3Department of Infection Prevention and Infectious Diseases, University Hospital Regensburg, 93053 Regensburg, Germany; wulf.schneider@klinik.uni-regensburg.de (W.S.-B.); baerbel.kieninger@klinik.uni-regensburg.de (B.K.); anja.eichner@klinik.uni-regensburg.de (A.E.); anca.rath@klinik.uni-regensburg.de (A.R.); juergen.fritsch@klinik.uni-regensburg.de (J.F.); 4fzmb GmbH, Forschungszentrum für Medizintechnik und Biotechnologie, 99947 Bad Langensalza, Germany; dgary@fzmb.de (D.G.); kfrankenfeld@fzmb.de (K.F.); twellhoefer@fzmb.de (T.W.); 5Institute of Physical Chemistry, Friedrich-Schiller University, 07743 Jena, Germany

**Keywords:** microarrays, enterococci, VRE, DNA, resistance genes, virulence genes, Gram-positive bacteria

## Abstract

Today, there is a continuous worldwide battle against antimicrobial resistance (AMR) and that includes vancomycin-resistant enterococci (VRE). Methods that can adequately and quickly detect transmission chains in outbreaks are needed to trace and manage this problem fast and cost-effectively. In this study, DNA-microarray-based technology was developed for this purpose. It commenced with the bioinformatic design of specific oligonucleotide sequences to obtain amplification primers and hybridization probes. Microarrays were manufactured using these synthesized oligonucleotides. A highly parallel and stringent labeling and hybridization protocol was developed and employed using isolated genomic DNA from previously sequenced (referenced) clinical VRE strains for optimal sensitivity and specificity. Microarray results showed the detection of virulence, resistance, and species-specific genes in the VRE strains. Theoretical predictions of the microarray results were also derived from the sequences of the same VRE strain and were compared to array results while optimizing protocols until the microarray result and theoretical predictions were a match. The study concludes that DNA microarray technology can be used to quickly, accurately, and economically detect specifically and massively parallel target genes in enterococci.

## 1. Introduction

Enterococci are facultative, anaerobic, Gram-positive bacteria that generally live as commensals in the gut of various organisms, including humans. Besides *Staphylococcus aureus* and certain Gram-negatives, they are also one of the leading causes of nosocomial infections like intra-abdominal infections, pelvic and wound infections (especially post-surgery), bacteremia, endocarditis, and urinary tract infections [1]. Most nosocomial infections with multidrug-resistant organisms are endogenous [2], and antimicrobial resistance is an increasing dilemma in healthcare as, particularly, vancomycin-resistant enterococci (VRE) are complicated to treat [3].

Most enterococcal infections in humans are caused by *Enterococcus faecalis* or *Enterococcus faecium.* Although they are opportunistic pathogens that become only relevant in the presence of certain risk factors, enterococci carry several virulence factors that help to establish an infection in the host organism [4]. These include, among others, *acm*—surface exposed antigen, *eep*—protease gene, *efaA*—adhesin (a solute binding-protein receptor for a manganese transport system in *E. faecalis* [5], *efmM*—methyltransferase [6], *fss3*—fibrinogen binding gene, *srtA2*—sortase gene, and *tirE1*—Toll/interleukin-1 receptor [7].

All enterococci are resistant to oxacillin, clindamycin, and cephalosporins. *E. faecalis* and *E. faecium* used to be susceptible to ampicillin, piperacillin, and imipenem, or to a combination of these beta-lactams with aminoglycosides. However, most clinical *E. faecium* are now resistant to aminopenicillins, while *E. faecalis* resistance to them is uncommon [2]. Mutations in *pbp5* (the gene encoding a penicillin binding protein) are associated with ampicillin resistance. The increase in the cases of ampicillin resistance in VRE can be as a result of changes in *pbp5* [1]. *Pbp5* is characterized as a low-affinity transpeptidase whose function is cell wall synthesis, and it is not easily susceptible to inhibition by beta-lactam antibiotics, such as penicillin. *pbp5* facilitates the cross-linking of peptidoglycan in the bacterial cell wall, thus preserving the structural integrity of the bacteria and enabling survival against antibiotic treatment. Variants of *pbp5*, particularly in *Enterococcus faecium*, can exhibit mutations that further decrease their affinity for penicillin. These mutations are linked to elevated levels of resistance and can arise during prolonged exposure to beta-lactam antibiotics [8]. Aminoglycosides alone are not used to treat enterococcal infections because of a low-level resistance. However, due to synergism with beta-lactams, they can be used together with compounds such as ampicillin or imipenem unless certain genes for aminoglycoside-modifying enzymes are present that confer a high-level resistance in which such combination therapy is not effective. It has been shown [9,10] that high-level-resistance aminoglycoside genes are encoded on plasmids or conjugative transposons, facilitating the transfer of resistance elements in enterococci. Common aminoglycoside-resistance genes that can be present in VRE, as well as in other bacteria, are *aad6* and *aacA*-*aphD.* Another aminoglycoside-resistance gene, *aac6* (GenBank number: L12710.1[169..717]), is a species-specific marker for *E. faecium* [11], while other AAC’6 (aminoglycoside 6′-N-acetyltransferase) proteins are encoded by genes also present in other bacteria [12].

Other resistance genes in enterococci include the tetracycline resistance genes *tetS* and *tetM*, as well as *ermB*, which is the most common macrolide/erythromycin-resistance gene in enterococci. Its gene product is a methylase that modifies the target site of the antibiotic and thus prevents the antibiotic from binding to its ribosomal target [13].

Glycopeptides (vancomycin and teicoplanin) inhibit the cell wall synthesis of Gram-positive bacteria as a result of the formation of complexes between the antibiotics and the carboxy-terminal D-alanine residues of peptidoglycan precursors [14]. Glycopeptides are important although poor tissue diffusion might pose a limitation to the therapy of other conditions than bloodstream infections. The genes *vanA*, *vanB*, *vanYB*, *vanS*, *vanZ*, and *vanH* represent glycopeptide-resistance genes. Acquired glycopeptide resistance mediated by *vanA* and *vanB* occurs as a result of both *van* operons encoding a ligase that alters the target binding site of vancomycin/glycopeptide, which is important for the antibiotics’ function. *vanA* and *vanB* are also the most common of all the *van* genes in enterococci [15]. VRE strains with *vanA* are resistant to both vancomycin and teicoplanin. The *vanB*-type strains are susceptible to teicoplanin [16].

Only a few therapeutic options remain in the case of vancomycin/glycopeptide resistance. These include quinupristin and dalfopristin (to which *E. faecalis* is commonly resistant), linezolid, tigecycline, and daptomycin [17,18].

The limited options for therapy require preventive measures against VRE infection and transmission. This includes molecular typing methods to be able to accurately detect transmission chains in outbreak situations and to distinguish endogenous infections from those transmitted within a hospital in order to facilitate adequate infection control, as well as to perform epidemiological studies if/when the need arises [19].

Pulse-field gel electrophoresis (PFGE) could be used as a typing method, and, although the method has been standardized, it is cumbersome and cannot easily be reproduced outside the reference laboratory. More so, extracting and analyzing relevant information from large datasets obtained and inter-laboratory comparison over time are challenging [20,21,22].

Multilocus sequence typing (MLST), which uses the principle of identifying nucleotide sequences in PCR products of multiple (usually seven) housekeeping genes, has been the “gold standard” for typing. The traditional MLST scheme for *Enterococcus faecium* was developed in 2002 [23] based on the *Enterococcus faecium* gene sequences and putative gene functions available at the time. However, the resolution of MLST is limited because of the small number of genes (just seven) used in the scheme [24]. Also, obtaining information on resistance and virulence genes within the typed strains would require more PCR runs because the MLST scheme provides only information on the seven phylogenic markers. Furthermore, the traditional procedure for MLST is time consuming and labor-intensive. There can be errors stemming from DNA polymerase used in polymerase chain reactions that could lead to mutation in produced amplicons [25]. The promotion of whole genome sequencing (WGS) has brought to light inconsistencies between the whole genome sequence data and traditional MLST [26,27,28,29].

A new MLST scheme (Bezdíček MLST) was recently proposed [26] based on the WGS data of strains collected within the Czech Republic and the worldwide *E. faecium* population whose genome sequences were obtained from different genome databases reflecting the individual sequence types true genetic relatedness as much as possible. Isolates that differ in a greater number of alleles are also grouped in this scheme [26]. Although the Bezdíček MLST scheme may be different from traditional MLST, it has similar limitations as the traditional MLST.

Core genome multilocus sequence typing (cgMLST) is another molecular typing method that uses core genomes to determine genetic relationships between strains, their population structure and genetic evolution [22]. The cgMLST technique uses WGS data that contain all of the genetic information that a strain encodes in evaluating genes of core genomes and allelic profile variations [30,31]. This technique is essentially a one-by-one comparison of all genes in a genome sequence instead of only seven housekeeping genes as in traditional MLST [24,32]. Compared to both, the traditional and the Bezdíček MLST schemes, cgMLST is more accurate and can distinguish differences between strains within the same MLST sequence types [26,33,34,35]. One disadvantage of WGS is that sequencing strains from multiple cases, such as during an outbreak, results in a very large data set that needs to be processed and handled to obtain a practically manageable, condensed result. Another disadvantage is the limited availability of the method, its costs, and its need for qualified workforce with the required technical know-how to operate, analyze, and correctly interpret results [30,36].

The present study focuses on fast and economical DNA-microarray-based technology. Microarray technology has come a long way today from its previous use, having its origins in the late 1980s, with the development of earlier techniques like dot blots and spotted arrays on nylon membranes. The development of microarray technology was a gradual process over the past few decades, with significant advancements in the 2000s driven by the availability of genomic sequence information and the need for high-throughput gene expression analysis tools. The transition from spotted arrays to inkjet-based fabrication methods was a key technical advancement that improved the quality and density of microarray platforms. Over the past decade, microarray technology has developed rapidly in terms of the number of features on an array and the statistical methods for analyzing the results. The diverse platforms and technical aspects of microarray manufacturing can be classified into two broad categories: those using pre-synthesized DNA sequences and those using in situ synthesis methods. The early microarray platforms were predominantly based on spotted arrays on glass slides, which had some quality control issues like non-uniform feature sizes and irregular shapes. The development of inkjet deposition methods for array fabrication helped overcome these quality issues and allowed for much higher feature densities, up to 185,000 features per slide. The flexibility of the microarray design, allowing for the inclusion of various types of features like cDNA, predicted genes, and intergenic regions, greatly increased the number of biological questions that could be addressed using this technology. The sequencing of the human genome in the early 2000s was a key trigger that propelled the development and widespread adoption of microarray technologies as they enabled researchers to study global gene expression changes in a high-throughput manner [37,38,39]. The technology was originally designed for the measurement of transcriptional levels of RNA transcripts obtained from a large number of genes within a genome in only one experiment [40].

Nowadays, as will be seen in this study, it is possible to immobilize probes for specific target genes on the DNA microarray. Due to the ability of complementary single-stranded sequences of nucleic acids to form double-stranded hybrids, it is possible to do primer extension reactions that label amplicons with biotin and the labeled sample DNA binds to its specific probe on the microarray. This is followed by a conjugation step that allows for the incorporated label to trigger dye precipitation at the DNA–DNA-hybridized probes, leading to spots on the array. This way, we can tell if a target gene is present when there is a spot or absent when there is none. More than 300 probes can be immobilized on the microarray, as was carried out for *Staphylococcus aureus* in previous studies [41,42]. Our aim was to include resistance, virulence, species-specific, and typing markers for VRE on microarrays for use as everyday affordable and disposable consumables in diagnostics and epidemiology as was already previously demonstrated for staphylococci [41,42,43,44].

## 2. Materials and Methods

### 2.1. Bioinformatic Analysis: Target Gene Selection, and Oligonucleotide Sequence Design

The first step was a comprehensive bioinformatic analysis in which high-quality annotated sequence data of enterococcal strains were collected from the Pathosystems Resource Integration Center (PATRIC) database—www.patricbrc.org (accessed on 21 September 2021). An in-house homology database was then created based on mmseq2 clustering [45]. Due to the reduced likelihood of mutations in regions that are under higher selective pressure, we defined consensus primer and probes for the assay based on multiple sequence alignments. The primers and probes were designed using the ConsensusPrime pipeline, where biophysical parameters like the length of their sequences, melting temperatures, and GC content were considered (see Appendix A) [46,47].

The resulting oligonucleotides were then ordered and synthesized from Metabion International AG (Planegg, Germany), and the microarrays were manufactured by INTER-ARRAY part of fzmb GmbH, Forschungszentrum für Medizintechnik und Biotechnologie (Bad Langensalza, Germany).

### 2.2. Strains/Isolates

One hundred and eighty previously Illumina-sequenced clinical vancomycin-resistant *E. faecium* strains were obtained from the University Hospital in Regensburg, Germany, and another two came from the University Hospital in Jena, Germany. In addition, one vancomycin-resistant *Enterococcus faecalis*, also from Jena, was included as a control.

### 2.3. Genome Sequencing

DNA extraction for WGS was performed with a QIAmp DNA Mini Kit (DNeasy Blood and Tissue Kit (250) QIAGEN, Hilden, Germany). DNA concentration and quality were measured by Qubit (dsDNA HS array kit, Thermo Fisher Scientific, Dreieich, Germany). Sequencing libraries were generated using the Nextera XT library Prep Kit (Illumina, San Diego, CA, USA), and sequencing was performed on either MiniSeq or NextSeq Dx550 (Illumina) with a 2 × 150 bp paired-end sequencing run using either a high-output (MiniSeq) or a mid-output cassette (NextSeq Dx550), respectively. The assembly with SKESA and the genome comparison with MLST and cgMLST were performed using SeqSphere+ (Ridom, Munster, Germany, Version 9.0.1) [48,49].

### 2.4. Genomic DNA Isolation from Enterococci

The next task was the development of an effective protocol for isolating high-quality, RNA-free, and unfragmented genomic DNA from different enterococcal strains resulting in the following protocol.

Two blood agar plates (BD, Heidelberg, Germany) of each enterococcal strain were cultured for 24 h at 37 °C to ensure a sufficient amount of DNA. A lysis buffer was used that consisted of 160 µL of a commercial lysis buffer (DNeasy Blood and Tissue Kit (250) QIAGEN, Hilden, Germany), 20 µL of a lysozyme stock solution (10 mg lysozyme by Sigma-Aldrich, Taufkirchen, Germany in 1 mL of phosphate-buffered saline, PBS; Invitrogen Fisher Scientific, Schwerte, Germany), 10 µL of an achromopeptidase stock solution (100 units achromopeptidase by Sigma-Aldrich in 5 mL of PBS), and 20 µL of Ribonuclease A solution (10 mg Ribonuclease A by Sigma-Aldrich in 1 mL of PBS). Two full 5 µL inoculation loops of the cultured enterococcal strain were added to the Eppendorf tube, vortexed, and incubated for 30 min at 37 °C and 550 rpm on a BioShake iQ shaker (QINSTRUMENTS, Jena, Germany) for optimal mixing (QINSTRUMENTS, Jena, Germany). After this step, 25 μL Proteinase K + 200 µL Buffer AL (DNeasy Blood and Tissue Kit (250) QIAGEN, Hilden, Germany) was added, vortexed, and incubated at 56 °C for 60 min at 550 rpm on a BioShake iQ shaker (QINSTRUMENTS, Jena, Germany).

The next step involved the addition of 200 µL of ethanol (96–100%), after which the sample mix was vortexed again. Then, the sample was transferred into the DNeasy Mini spin column (DNeasy Blood and Tissue Kit (250) QIAGEN, Hilden, Germany) and centrifuged for 1 min at 8000 rpm. The column run tube was discarded, and 500 µL buffer AW1 (DNeasy Blood and Tissue Kit (250) QIAGEN, Hilden, Germany) was added to the DNeasy spin column. It was placed in a new run tube, centrifuged for 1 min at 8000 rpm, and the run tube was again discarded. The spin column was put afresh in a new run tube, and 500 µL buffer AW2 (DNeasy Blood and Tissue Kit (250) QIAGEN, Hilden, Germany) was added and centrifuged for 3 min at 14,000 rpm to ensure that the spin column was dry and free of remnants of wash buffers. The run tube was removed and discarded, and the spin column was placed in a 1.5 mL Eppendorf tube. Then, 100 µL elution buffer AE (DNeasy Blood and Tissue Kit (250) QIAGEN, Hilden, Germany) was added and incubated for 1 min at room temperature; the tube was then centrifuged for 1 min at 8000 rpm. This elution step was repeated again, increasing the total volume of eluate to 200 µL. An Eppendorf concentrator-plus device (Wesseling-Berzdorf, Germany) was used to condense the DNA volume by half at 45 °C for 50 min.

### 2.5. Microarray Production

The microarrays (INTER-ARRAY part of fzmb GmbH, Forschungszentrum für Medizintechnik und Biotechnologie (Bad Langensalza, Germany) were produced in two batches as in Table 1, to stepwisely increase the number of target genes once procedures had been established (see Appendix A).

The microarrays were produced in strips (8 microarrays in one strip). This allows for the possibility of at least 8 microarray experiments being run simultaneously.

### 2.6. Labeling and Hybridization

INTER-ARRAY Labeling and Hybridization Kits for VRE contained 3-0 ArrayWell strips VRE, 2-3 Primer Mix VRE (synthesized by Metabion as described above and prepared as a ready-to-use mix by INTER-ARRAY part of fzmb GmbH, Forschungszentrum für Medizintechnik und Biotechnologie, Bad Langensalza, Germany), 2-0 Labeling buffer (includes biotin dUTP), 2-2 Labeling enzyme, 3-1 Hybridization Buffer, 3-2 Washing Buffer 1, 4-1 HRP (streptavidin–horseradish peroxidase) Conjugate (100×), 4-2 Conjugate Buffer, 4-3 Washing Buffer 2, and 5-1 HRP Substrate Solution blue.

Next, 3.9 µL 2-0 Labeling buffer (includes biotin dUTP), 0.1 µL 2-2 Labeling enzyme, and 1 µL 2-3 Primer Mix VRE were added to a 1.5 mL Eppendorf tube vortexed and transferred to a 0.2 mL PCR tube (ThermoFisher Scientific, Bremen, Germany), to which 5 µL unfragmented DNA from VRE isolates were then added, vortexed, and shortly centrifuged. A PCR thermal cycling device (FlexCycler, Analytik, Jena, Germany) was used to perform the linear amplification and site-specific labeling process of the single-stranded DNAs. A linear primer extension reaction labeled amplicons by including biotin-dUTP. The conditions were an initial denaturation for 5 min at 96 °C, followed by 45 cycles of 60 s at 96 °C, 20 s at 62 °C, and 40 s at 72 °C.

The stringent hybridization steps were performed using a BioShake iQ, and the procedure included, first, pre-washing of the microarray with 150 µL distilled water at 50 °C, 5 min, 550 rpm. After this, just as after every step, leftover fluids were pipetted out using Pasteur pipettes, avoiding contact with the microarray surface. Then, a second washing step took place with 150 µL 3-1 Hybridization Buffer at 50 °C, 5 min, 550 rpm. The hybridization step commenced with the addition of 90 µL 3-1 Hybridization Buffer to a 10 µL labeled DNA sample. Both combined were added to the microarray tube, which was then closed and incubated at 50 °C for 60 min at 550 rpm. Afterwards, a washing step followed to ensure the array surface was clear of residual DNA and hybridization buffer. Then, 150 µL 3-2 Washing Buffer 1 was added to the microarray and withdrawn back and forth three times before pipetting all out. One hundred and fifty microliters of 3-2 Washing Buffer 1 was added again at 40 °C, 10 min, 550 rpm.

The blocking and conjugation steps that ensured adhesion of the detection enzyme to only the points where DNA–DNA hybridization had happened on the microarray were next. Ninety-nine microliters of 4-2 Conjugation Buffer combined with 1 µL 4-1 HRP Conjugate was added to the microarray tube and incubated at 30 °C, 10 min, 550 rpm. A final washing step was carried out with 150 µL 4-3 Washing Buffer 2 added to the array and withdrawn back and forth three times before pipetting it all out. Another 150 µL 4-3 Washing Buffer 2 was added to the microarray tube at 30 °C, 2 min, 550 rpm.

The staining step was performed by the addition of 100 µL 5-1 HRP Substrate Solution blue to the microarray at room temperature and incubated at 25 °C for 6–8 min, without mixing or shaking. The incorporated HRP conjugate bound with its substrate triggered dye precipitation at points where DNA–DNA hybridization had occurred on the microarray, leading to the formation of visible spots on the array surface.

### 2.7. Data Analysis

The image of the microarray was taken by a dedicated reader and software—INTERVISION GENOTYPING 1.1.0 (INTER-ARRAY part of fzmb GmbH, Bad Langensalza, Germany). The microarray layout we had previously designed (number, arrangement, repetitions of the same substances, and concentrations (see Appendix A)) was provided to the device’s software by an assay-specific plug-in file to adapt it to this particular assay. The microarray layout is a fixed coordinate grid with position markers allowing us to determine which targets were present or absent based on the pattern of spots on the surface of the microarray. The INTER-ARRAY reader can process up to 12 microarray strips in one run.

The genome sequences of the clinical VRE strains were obtained, and primer and probe sequences were mapped to the genome sequences of each individual strain. These theoretical results were then compared to the experimental results to assess complete concordance via the modification of the stringency of the hybridization protocol. The purpose of this was not only to have a proof of concept but to assess the appearance of hybridized and stained microarray strips in order to develop the protocols for the experiments and determine the hybridization temperatures and buffer selection. It also served as quality control for the microarray experiments.

The receiver operating characteristic (ROC) curve, which is the plot of sensitivity against specificity and the area under the curve (AUC) [50], was used to evaluate the performance of the primer/probe pair for each target in all samples when compared to these theoretical results (see Appendix A). In order to obtain an optimum threshold value for discriminating between positive and negative spots, outliers were first removed. All measured values within a group outside 2.5 times the standard deviation were defined as outliers. It was then iterated in steps of 0.01 over all possible threshold values of the normalized grey values. The threshold value that best separates the two groups is the so-called “Step_Threshold”. If several threshold values are equally good, their average was used. It should be noted that threshold values are only statistically robust for equally distributed data sets based on more genetically diverse strains, which is not the focus of this paper.

### 2.8. Analyzing Similar Hybridization Profiles for Typing Purposes

The similarity or identity of isolates can also be assessed visually on small scales. However, when dealing with higher numbers of isolates, hybridization profiles of the isolates from the microarray were converted into “sequences” that can be analyzed using SplitsTree 4 software [51] using default settings (characters transformation, uncorrected P; distance transformation, Neighbour-Net; and variance, ordinary least squares) [52,53] in order to visualize clusters of isolates with similar or identical hybridization profiles. It should be noted that this does not necessarily imply a true phylogenetic relationship (because many of the genes included are situated on mobile genetic elements) but serves the purpose of assessing the similarity of profiles in a reproducible, user-independent way.

## 3. Results

### 3.1. Optimization of DNA Isolation Protocol

Repeated DNA isolation experiments using the DNeasy Blood and Tissue Kit (250) lead to the introduction of RNase, lysosome, and achromopeptidase as lysing agents just as described in the Section 2 in order to achieve the desired unfragmented RNA-free DNA concentration of at least 30 ng/mL for subsequent labeling and hybridization steps. Some of the VRE strains had less colony growth than others and a watery consistency, leading to the streaking of each VRE strain on two blood agar plates to ensure enough colony material for genomic DNA isolation.

### 3.2. Comparison between Microarray Results and Sequence-Based Predictions

Genome sequences were known for all strains used in these verification experiments. Every probe (representing species markers, resistance, and virulence genes) immobilized on the microarray was searched within these sequences, resulting in a “theoretical experiment” (i.e., the sequence-based predictions as described above), which predicted the pattern of the microarray derived from the sequences. The comparison between in silico and in vitro experiments was a first proof of concept and served as quality control for the entire methodological concept as well as for array production, although for multiple isolates, this was carried out using a spreadsheet editor (excel 2019) rather than images as Figure 1 (see Appendix A). *E. faecium* species markers were present in 182 strains, and *E. faecalis* species markers were present in one strain. These results were in complete alignment with the theoretical experiments. They confirm concordance between the computerized predictions and the microarray experiments, proving that the microarray works as it should.

Additionally, a ROC curve analysis for 78 target genes in 183 samples was performed. The area under the curve (AUC) for 71 of these targets was 1, and others like *fexB*, *pbp5_2*, *pbp5_3*, *vanS-G*, *bepA_fruA*, *pilF*, and *prpA* had AUCs of 0.84, 0.88, 0.97, 0.75, 0.99, 0.85, and 0.69, respectively.

Diagnostic sensitivity and specificity in 72 of the 78 targets were both 100%. *fexB*, *pbp5_2*, *pbp5_3*, *vanS-G*, *pilF*, and *prpA* had sensitivities of 100%, 70%, 100%, 75%, 60%, and 56% and a specificity of 66%, 100%, 94%, 100%, 100%, and 79%, respectively.

Fifty-eighty of the targets, namely, vanT_C1, vanT_C234, vanU, vanSFM, vatD, vatE, vgbA, Egal_vanC1, every other vanC genes, ace, bee1, bee2, bee 3, vanYD, vanXY, tetK, aad6, aadA, cfr2, cfrB, ermA, fexA, optrA, vanRG, vanSCE, and asp1 were excluded because they were absent from all the strains, and aac6, srtA1, srtA2, and sagA faecium were also excluded because they were present in all the strains used in this experiment and so were not suitable for ROC analysis. Species markers were not yet included in the ROC analysis due to insufficient data, as *E. raffinosus*, *E. hirae*, *E. casseliflavus*, *E. durans*, and *E. avium* strains were not included in this study. Negative control microarray experiments without sample DNA were also excluded from the ROC analysis (see Appendix A).

An overview of the theoretical predictions, microarray experiments, and ROC analysis can be seen in Appendix A. Step thresholds for the separation of negative and positive values for all data sets ranged from 0.27 to 0.64, with the most common thresholds being between 0.40 and 0.64.

### 3.3. Detection of Resistance and Virulence-Associated Genes Alongside Species Markers

Regarding resistance markers, there were 90 isolates positive for *vanB* and 41 isolates positive for the presence of *vanA* genes (see Figure 2), while *vanC* was not present in any of the isolates in this experiment. Other van operons like *vanH*, *vanYB*, *vanR*, and *vanZ* appeared in 107, 132, 55, and 140 isolates, respectively (see Table 2). A total of 125 strains were positive for the *tetM* gene, and two strains were positive for *poxtA*; one strain had a *qnr* gene present, and *tetK* and *fexA*, like *vanC*, were not present in all strains. The virulence genes *acm*, *eep*, *bepA*, *efbA*, *efmM*, and *srtA2C* could be detected in 113, 138, 182, 135, 123, and 94 strains, while one strain was positive for *lsaA* and *ace.* The species markers *ddlA_faecium*, *dnaA*, *tuf*, *sodA*, *rpmB*, and *recG* for *E. faecium* were found in the 182 isolates previously identified as *E. faecium*, and *ddl_faecalis*, *dnaA*, *tuf*, *sodA*, *rpmB*, and *recG* for *E. faecalis* were found in the one *E. faecalis* strain. These results are also provided in Appendix A.

### 3.4. Generation of Genetic Fingerprints for Typing Purposes

Strains with identical hybridization profiles on the second-generation microarray (see Appendix A) were sub-grouped in eighteen different clusters. Table 3 and more detailed Appendix A show an overview of the VRE clusters comprising strains with identical hybridization patterns that consisted of 36, 16, 7, 5, 3, and 2 strains, respectively. Each of the remaining 74 strains had unique hybridization patterns.

The corresponding MLST sequence types according to both schemes, and complex type IDs, are also provided, showing that the array data provided a resolution similar to CT typing and a better one than either MLST scheme.

Using the traditional MLST scheme, sixty-nine of the VRE strains were ST117, fifty-three strains were ST80, fourteen were ST1299, others were ST78, ST262, ST192, ST272, ST375, ST203, ST721, ST17, ST18, and ST25, and five of the strains had no known ST.

Most of the strains in a group were ST17, ST21, and ST18, according to the Bezdíček MLST scheme [26,56]. Others were ST57, ST123, ST143, ST166, ST318, ST327, and ST504.

With cgMLST, some of the isolates in a cluster had complex type (CT): 71, 1065, and 1470. Other CTs are 469, 929, 1573, 1903, 2505, 7050, 7081, 2967, 3292, 6002, 2035, 7163, and 7074 [48,49].

The sequences derived from the hybridization profiles were used as input for visualization to show the clusters of isolates (see Figure 3) with identical hybridization profiles [36].

## 4. Discussion

Antimicrobial resistance, especially as seen in VRE, continues to be a problem that needs to be tackled on all fronts [57] to effectively combat the growing threat. Implementing stringent infection control measures, such as hand hygiene, contact precautions, and environmental disinfection, is essential to prevent the spread of VRE in healthcare settings. There is a dire need for (amongst others) the development of fast, cost-effective, and standardized molecular multiparameter analyses for diagnostic and epidemiological tools that can be manufactured and used as everyday consumables in diagnostic and surveillance laboratories. DNA microarray technology gives us this opportunity, as it can be adapted to new markers and strains, and the array results are easily comparable between different parts of the world.

The merit of this study was to develop a DNA microarray designated for VRE. This included the establishment of sets of primers and probes that were used in the production of the VRE microarrays and the optimization of protocols for procedures like the isolation of genomic DNA in enterococci, labeling, and hybridization, as well as the development of procedures for the verification of the microarray results. Parameters like temperature, buffer concentration, incubation time, and additional lysing agents in specific quantities were optimized until in silico simulated results were accurately reproduced in vitro on the microarray. Therefore, with the resulting VRE microarray, resistance, virulence, and species-specific markers were detected accurately, and typing results were reasonable, i.e., comparable to CT typing. There are 139 targets on the VRE microarray (VRE-02), including spotting buffers as negative and biotin markers as positive staining controls, 64 resistance markers, 47 virulence markers, and 25 species markers.

A comparison among clusters of identical hybridization profiles on the microarray and existing typing schemes shows that the clusters match better with cgMLST and, in a few cases, with the Bezdíček MLST than with the traditional MLST. Some strains used in the study are part of the rapidly emerging *vanA*-positive ST1299, endemic to the southern part of Germany [49].

These strains are also mostly CT1903 and CT3109 (see Table 3). The ST1299 “clone” can be split into two clusters—Cluster 5 and 6—based on their hybridization patterns (array patterns RGB-[D16_S94] and RGB-[D1855_S64]) on the microarray allowing us to prove or disprove cases of transmission in an outbreak of this emerging ST1299 “clone” without the need to repeat WGS in every case.

Whether strains with identical or similar hybridization profiles are indeed phylogenetically linked should be established by studying genome sequences of more diverse strains. However, some strains with identical/similar hybridization profiles were epidemiologically connected because these samples were obtained from in-patients in the same hospital ward who were admitted within a close period. Although the link between the remaining isolates in the same group could not be established, that does not necessarily translate into an absence of a connection among them, but rather a difficulty in establishing these links outside the hospital settings.

As mentioned already, the genotyping of bacteria, particularly enterococci, using highly parallel microarray-based assays offers significant advantages over traditional molecular methods such as sequencing, MLST, cg-MLST, and various PCR techniques in the context of antimicrobial resistance. One of the primary benefits of microarray assays is their ability to simultaneously detect multiple resistance, virulence, species, and typing genes in a single test, thereby significantly enhancing throughput and efficiency. This multiplex capability reduces the overall workload compared to methods like PCR, which require separate reactions for each target gene, thus streamlining laboratory processes and minimizing the time from sample to result. Additionally, microarray assays typically demonstrate high sensitivity and specificity, allowing for accurately identifying resistance determinants, which is critical for effective clinical decision-making. In terms of cost, microarray technology can be more economical when considering the comprehensive data it provides in one assay, as opposed to the cumulative costs of multiple tests needed for other methods.

Furthermore, microarrays are generally more accessible and easier to implement in routine laboratory settings due to their standardized protocols and reduced need for specialized equipment compared to WGS, which, while powerful, often requires extensive bioinformatics support and can be more expensive and time-consuming. The reproducibility of microarray assays is another advantage, as they provide consistent results across different runs and laboratories, which is essential for epidemiological studies and surveillance programs. In contrast, results from methods like MLST and cg-MLST can be more variable due to differences in allele calling and the interpretation of sequence data.

Moreover, microarray assays can be rapidly adapted to include new resistance genes as they are identified, ensuring that the detection capabilities remain current in the face of evolving AMR patterns. This adaptability is crucial for public health responses to emerging threats. Overall, using highly parallel microarray-based assays for genotyping enterococci represents a robust, efficient, and cost-effective approach for monitoring AMR, facilitating timely interventions and enhancing our understanding of resistance mechanisms in clinical settings. A wise strategy for typing would also be screening isolates with microarrays, and, based on the results, a decision is made about which isolates still require WGS. The reasons for WGS in these situations could be atypical characteristics of a new strain and/or indistinguishable isolates. With the microarray, the non-identical isolates could be easily sorted out, thereby saving some cost and effort that would have been required for WGS. The identical strains can presumably be handled as an outbreak, i.e., an infection control intervention could be rapidly triggered based on acceptable evidence.

The primary objective of this study was to establish the DNA microarray technique for VRE. While this goal was successfully achieved, certain limitations were acknowledged and will be addressed in subsequent studies. One of these limitations is the testing of only clinical VRE strains, mostly obtained from one region in the southern part of Germany. A substantial amount of VRE strains from different parts of the world still need to be tested to build a robust database for VRE hybridization profiles. Another limitation is the exclusion of other enterococcal species like *E. hirae*, *E. casseliflavus*, *E. durans*, and *E. gallinarium,* etc. These enterococcal species should be tested with the VRE microarray because that would assess the functionality of the species markers. Lastly, the target panel should be expanded, including more target genes that should serve the purpose of typing. This is important because an increase in the number of putative typing markers on the microarray will increase the resolution of typing.

In summary, the VRE microarray developed herein detects resistance, virulence, and species markers accurately. It has also shown reasonable typing capability, which will be expanded further by including additional typing markers in the next phase.

## Figures and Tables

**Figure 1 sensors-24-06476-f001:**
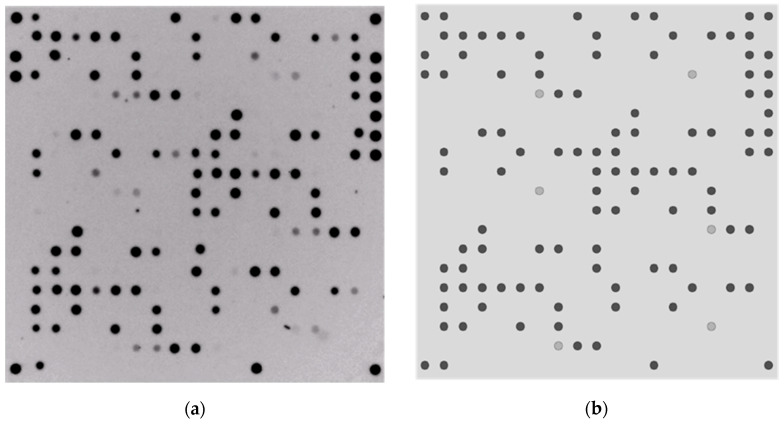
(**a**) On the left is a picture of the microarray for ST117 strain “Regensburg 105-008” following its genomic DNA isolation, labeling, and hybridization steps. The spots are representative of all the target genes present in the strain. (**b**) On the right is a picture obtained from the simulation of the experiment using the sequence data of the same strain. The probes representing each target gene were searched within the genome sequence. Dark grey spots indicate perfectly matching probes, and light-grey spots represent probes with two mismatches to the target sequence. A visual comparison of the two pictures shows concordance.

**Figure 2 sensors-24-06476-f002:**
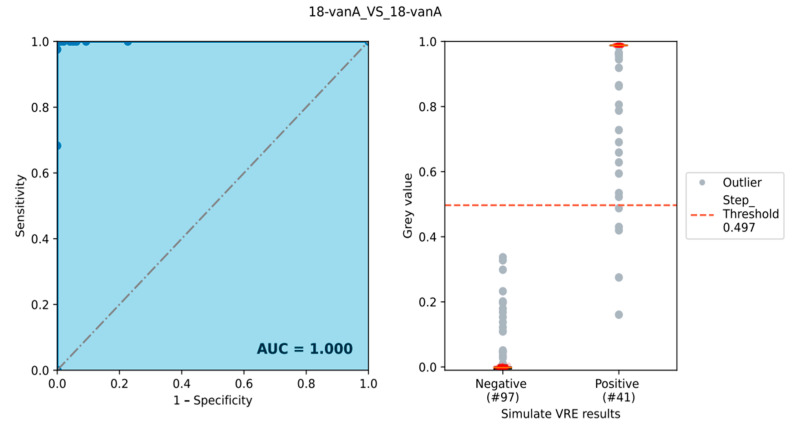
ROC curve analysis showing AUC of 1.000 and a box plot showing the step threshold of 0.497 for separation of *vanA* positive and negative samples on the array. The gray dots were classified as outliers and not considered in the threshold prediction.

**Figure 3 sensors-24-06476-f003:**
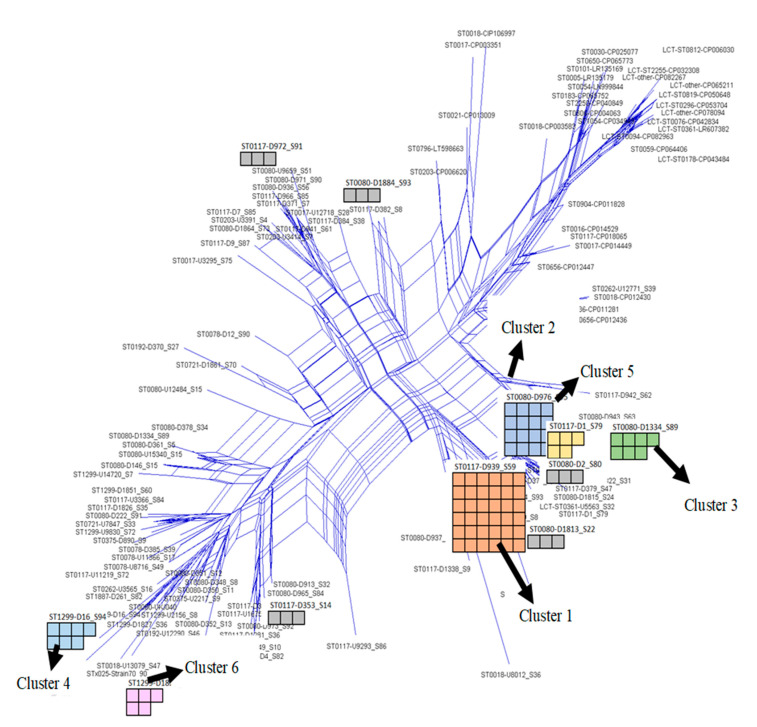
A similarity matrix (see Section 2) showing hybridization profiles of identical isolates in groups. Clusters 1, 2, 3, 4, 5, and 6 have 36, 16, 7, 7, 5, and 5 isolates in each group.

**Table 1 sensors-24-06476-t001:** Microarray batches, groups of target genes, and the total number of target probes immobilized on each array.

Microarray	Target Probes on the Array	Number of Target Probes on the Microarray
1st generation	Biotin, resistance, and virulence markers	111
2nd generation	Biotin, resistance, virulence, and species-specific markers	139

**Table 2 sensors-24-06476-t002:** The table shows a list of some of the target genes present on the microarray, the number of VRE strains in this study that have these targets present or absent in them, and the step threshold of each target and their ROC AUC post-ROC curve analysis (full list in Appendix A).

GenBank Number	Target Genes	Target Gene Description	No. of Positives	No. of Negatives	Step Threshold	* ROC AUC
** *Resistance markers* **						
ACAX01000144.1 [280:1017]	*ermB*	Macrolide resistance	144	12	0.5	1
ACBB01000372.1 [37:1476]	*aacA-aphD*	Aminoglycoside resistance	47	93	0.5	1
CP003583.1 [1443763:1445799]	*pbp5_3*	Penicillin-binding protein 5 (associated with ampicillin resistance) [8]	179	4	0.6	0.98
CP071931.1 [853137:854165]	*vanB*	Vancomycin resistance (van operon B)	90	66	0.64	1
GQ484956.1 [28460:29491]	*vanA*	D-alanine-D-lactate ligase/vancomycin resistance	41	97	0.5	1
CP068244.1 [894269:894844]	*vanZ*	Transmembrane teicoplanin-resistance protein (van operon Z)	140	1	0.49	1
CP014452.1 [34706:35674]	*vanH_1*	D-alanine-D-serine ligase/vancomycin resistance	107	1	0.49	1
AF162694.1 [3008:5104]	*vanC*	Membrane-bound serine racemase (glycopeptide resistance gene)	0	183	-	-
CP019989.1 [102852:103547]	*vanR*	Two component sensor/regulator, transcriptional regulator	55	101	0.5	1
CP003583.1 [1174985:1175809]	*vanYB*	D-Ala-D-Ala carboxypeptidase (glycopeptide resistance gene)	132	1	0.49	1
CP012454.1 [2804893:2806020]	*tetM*	Tetracycline resistance through ribosomal protection	125	29	0.5	1
CP006030.1 [2020974:2022182]	*qnr*	Quinolone resistance	1	159	0.5	1
KT892063.1 [105:2072]	*poxtA*	Linezolid resistance	2	160	0.55	1
CP066673.1 [1667698:1668903]	*tetK*	Tetracycline resistance	0	183	-	-
CP068244.1 [1940965:1942392]	*fexA*	Florfenicol resistance	0	183	-	-
AEBZ01000030.1 [377201:377944]	*lsa(A)*	Quinupristin + dalfopristin resistance intrinsic in *E. faecalis*	1	150	0.51	1
** *Virulence markers* **						
CP003583.1 [2228117:2230282]	*acm*	Collagen-binding microbial surface components recognizing adhesive matrix molecules (MSCRAMM) in *E. faecium*	113	1	0.40	1
CP018070.1 [73480:75456]	*pilA*	Major pilin subunit	62	98	0.61	1
AE016830.1 [537810:538529]	*efaA*	Adhesion protein plays role in endocarditis	109	6	0.49	1
CP012465.1 [759897:761699]	*bepA*	Permease	182	1	-	1
CP018071.1 [530776:534003]	*ecbA*	Collagen-binding MSCRAMM	118	19	0.5	1
CP012465.1 [1664985:1665647]	*eep*	Protease said to play a role in endocarditis [54,55]	138	1	0.48	1
CP012522.1 [252130:253836]	*efbA*	Adhesion protein plays role in endocarditis [54,55]	135	1	0.64	1
CP018065.1 [2272402:2273688]	*efmM*	Ribosomal RNA (rRNA) methyltransferase	123	1	0.48	1
CP006620.1 [2865948:2866424]	*esp*	Enterococcal surface protein	124	25	0.5	1
CP003351.1 [950876:951718]	*srtA2C*	Biofilm and pilus-associated sortase	94	34	0.5	1

* ROC AUC—receiver operating characteristic area under the curve.

**Table 3 sensors-24-06476-t003:** Clusters of VRE isolates that have similar microarray patterns, multilocus sequence types, a complex type, and the number of strains among each cluster with traceable epidemiological links.

Clusters	* MLST	Bezdíček MLST	CT	Array Patterns	No. of VRE Strains with the Same Array Pattern	No. of VRE Strains within the Cluster with Traceable Epidemiological Link
**1**	117/262	17/57	71/7077/1686/1917/1775	RGB-[D939_S59]	36	4
**2**	80/117	17/21	71/1065/3243	RGB-[D379_S47]	16	0
**3**	80/117	143/327/318	1473/7078/7059/2406/2403	RGB-[D1_S79]	7	0
**4**	80	504	1470	RGB-[D1334_S89]	7	0
**5**	1299	N/A	3109/1903	RGB-[D16_S94]	5	0
**6**	1299	N/A	1903	RGB-[D1855_S64]	3	0
**7**	117	18	2505/7074	RGB-[D353_S14]	3	0
**8**	0721	N/A	1573	RGB-[U7847_S33]	3	0
**9**	80	N/A	7050	RGB-[D1813_S22]	3	2
**10**	80	21	1065	RGB-[D1884_S93]	3	2
**11**	80	21	1065/7063	RGB-[D1860_S69]	3	0
**12**	-	18	929/2967/3292	RGB-[U3453_S12]	3	2
**13**	117	17/18	469/7081	RGB-[D941_S61]	2	0
**14**	80/117	143	7056/7060	RGB-[D1815_S24]	2	0
**15**	80	102	1565	RGB-[D0965_S84]	2	0
**16**	80	102	1565	RGB-[D973_S92]	2	0
**17**	117	166	6002	RGB-[D9_S87]	2	0
**18**	-	123/18	2035/7163	RGB-[D363_S21]	2	0

* MLST—multilocus sequence type, RGB—Regensburg, CT—complex type, and N/A—Unknown.

## Data Availability

The datasets generated for this study are available on request to the corresponding author.

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
