# Peer review of "Fast and Economic Microarray-Based Detection of Species-, Resistance-, and Virulence-Associated Genes in Clinical Strains of Vancomycin-Resistant Enterococci (VRE)"

_sensors, 2024, doi:10.3390/s24196476_

Round 1

Reviewer 1 Report

Comments and Suggestions for Authors

The manuscript provided a detailed description of the work and the results were quite impressive.  The reviewer appreciates the inclusion of the supplementary information. However, the description of the potential of this as a diagnostics needs more justification in terms of cost comparison and time to results to other existing technologies.  The overall time to get the results seems to be long.  It needs to get isolates, then extraction and linear amplification, followed by the DNA array and signal generational using HRP.  Because of that, it might not affect the treatment decisions or selection of antibiotics at the beginning.

The major weakness of this manuscript is the lack of image data on negative control samples or non-specific pathogens to show the specificity of the DNA spots on the detection.  It would be great if the authors can consider adding the info as supplemental information.

Author Response

Response to Reviewer 1 Comments

1. Summary

Thank you very much for taking the time to review this manuscript. Please find the detailed responses below and the corresponding revisions/corrections highlighted changes in the re-submitted files.

2. Questions for General Evaluation

Reviewer’s Evaluation

Response and Revisions

Does the introduction provide sufficient background and include all relevant references?

Yes

Are all the cited references relevant to the research?

Yes

Is the research design appropriate?

Yes

Are the methods adequately described?

Yes

Are the results clearly presented?

Can be improved

Are the conclusions supported by the results?

Yes

3. Point-by-point response to Comments and Suggestions for Authors

Comments 1:

The manuscript provided a detailed description of the work, and the results were quite impressive.  The reviewer appreciates the inclusion of the supplementary information. However, the description of the potential of this as a diagnostic needs more justification in terms of cost comparison and time to results to other existing technologies.  The overall time to get the results seems to be long.  It needs to get isolates, then extraction and linear amplification, followed by the DNA array and signal generational using HRP.  Because of that, it might not affect the treatment decisions or selection of antibiotics at the beginning.

Response 1: Thank you for your review. It is important to note that the molecular microarray-based multiplex VRE test is not an approved diagnostic test under CE-IVD regulations; rather, it serves as an epidemiological tool. This test efficiently assesses a wide range of molecular targets for their presence or absence, achieving results in a highly economical and relatively rapid manner—typically within one working day from culture to the fully automated evaluation of results. The process takes approximately 10 hours hands on time from sample to fully automated report generation. One of the key advantages of this method is its ability to provide simultaneous results for all targets present on the microarray. Additionally, multiple experiments can be conducted in parallel, as each microarray strip contains 8 wells, allowing for the analysis of 8 samples simultaneously. This parallel processing capability has been detailed on page 5, lines 248 and 249 of the manuscript. Moreover, the thermocycler used in this process can accommodate up to 96 wells, enabling multiple array strips containing samples for labeling (linear amplification) to be processed concurrently. The INTER-ARRAY imaging device can read up to 12 microarray strips in a single run, as mentioned on page 6, line 299-300. While PCR methods may offer faster results, they typically amplify only one target at a time. Even with multiplex PCR, the number of targets that can be analyzed does not match the extensive range available on the VRE microarray. Whole genome sequencing, as previously discussed in the manuscript, may take an equivalent amount of time or longer and requires careful analysis and interpretation of results. In outbreak situations where numerous samples need to be processed, the DNA microarray can be a valuable tool for typing. Regarding cost-effectiveness, the cost for a single microarray test is approximately €25, making it more affordable than other molecular typing methods. In summary, while the molecular microarray-based multiplex VRE test is not a CE-IVD approved diagnostic test, it offers a compelling alternative for epidemiological studies by efficiently analyzing multiple targets simultaneously at a lower cost and with comparable results to other methods like multiplex PCR and sequencing.

Comments 2:

The major weakness of this manuscript is the lack of image data on negative control samples or non-specific pathogens to show the specificity of the DNA spots on the detection.  It would be great if the authors can consider adding the info as supplemental information.

Response 2:

Thank you for your comment. We have now only negative control microarray experiments without DNA input and will add that as supplemental file D (see page 8, line 370 - 371, page 15, line 528 - 529).

Reviewer 2 Report

Comments and Suggestions for Authors

The manuscript describes microarray-based detection of resistance-, virulence- and species-associated genes in clinical strains of vancomycin resistant enterococci. The manuscript presented is detailed, interesting and well-organized. DNA microarrays are the promising tool for express and low-cost diagnostic devices development. I can recommend this manuscript for publication in Sensors journal after some minor changes:

1)      Please add the full information about the equipment and kits used, i.e. (Manufacturer, City, Country).

2)      Please add the information about buffer solutions’ content where it is possible.

Author Response

Response to Reviewer 2 Comments

1. Summary

2. Questions for General Evaluation

Reviewer’s Evaluation

Response and Revisions

Does the introduction provide sufficient background and include all relevant references?

Yes

Are all the cited references relevant to the research?

Yes

Is the research design appropriate?

Yes

Are the methods adequately described?

Yes

Are the results clearly presented?

Yes

Are the conclusions supported by the results?

Yes

3. Point-by-point response to Comments and Suggestions for Authors

Comments 1:

The manuscript describes microarray-based detection of resistance-, virulence- and species-associated genes in clinical strains of vancomycin resistant enterococci. The manuscript presented is detailed, interesting and well-organized. DNA microarrays are the promising tool for express and low-cost diagnostic devices development. I can recommend this manuscript for publication in Sensors journal after some minor changes:

Please add the full information about the equipment and kits used, i.e. (Manufacturer, City, Country).

Thank you for your comment. To address the first recommended change, we have added information about all equipment at the beginning of every method and only added the company subsequently to avoid repetition. However, I have changed it so that it shows in every case the manufacturer, city and country. This can be seen in page 4 line 195, page 5, lines 218, 220-222, 225, 227, 230, 233; page 6, lines 253 and 293.

Comments 2:

Please add the information about buffer solutions’ content where it is possible.

Response 2:

Regarding the buffer solutions used in the study, the specific compositions or buffer types are not explicitly stated for the commercial products from QIAGEN (utilized for genomic DNA isolation) or for the INTER-ARRAY system. However, the intended uses (e.g., for lysis, washing) and quantities of these buffer solutions have been specified in the manuscript. While the exact formulations of the buffer solutions are not provided, the information regarding their applications and amounts has been included in the relevant sections of the manuscript. This level of detail should enable readers to understand the overall workflow and procedures employed in the study, even without the precise buffer compositions being disclosed. If further clarification is needed regarding the buffer solutions, please let me know, and I will do my best to address any additional questions or concerns you may have.